# Citrus Auraptene Induces Glial Cell Line-Derived Neurotrophic Factor in C6 Cells

**DOI:** 10.3390/ijms21010253

**Published:** 2019-12-30

**Authors:** Yoshiko Furukawa, Ryu-ichi Hara, Makiko Nakaya, Satoshi Okuyama, Atsushi Sawamoto, Mitsunari Nakajima

**Affiliations:** Department of Pharmaceutical Pharmacology, College of Pharmaceutical Sciences, Matsuyama University, 4-2 Bunkyo-cho, Matsuyama 790-8578, Japan; 16140866@g.matsuyama-u.ac.jp (R.-i.H.); mu.yakuri.015@gmail.com (M.N.); sokuyama@g.matsuyama-u.ac.jp (S.O.); asawamot@g.matsuyama-u.ac.jp (A.S.); mnakajim@g.matsuyama-u.ac.jp (M.N.)

**Keywords:** auraptene, glial cell line-derived neurotrophic factor, GDNF, neuroprotective effect

## Abstract

We previously demonstrated that auraptene (AUR), a natural coumarin derived from citrus plants, exerts anti-inflammatory effects in the brain, resulting in neuroprotection in some mouse models of brain disorders. The present study showed that treatment with AUR significantly increased the release of glial cell line-derived neurotrophic factor (GDNF), in a dose- and time-dependent manner, by rat C6 glioma cells, which release was associated with increased expression of GDNF mRNA. These results suggest that AUR acted as a neuroprotective agent in the brain via not only its anti-inflammatory action but also its induction of neurotrophic factor. We also showed that (1) the AUR-induced GDNF production was inhibited by U0126, a specific inhibitor of mitogen-activated protein kinase/extracellular signal-regulated kinase (ERK) 1/2, and by H89, a specific inhibitor of protein kinase A (PKA); and (2) AUR induced the phosphorylation of cAMP response element-binding protein (CREB), a transcription factor located within the nucleus. These results suggest that AUR-stimulated *gdnf* gene expression was up-regulated through the PKA/ERK/CREB pathway in C6 cells.

## 1. Introduction

Auraptene (7-geranyloxycoumarin, AUR) is a coumarin derivative isolated from the peel of certain citrus fruits, and it is the most abundant form of naturally occurring geranyloxycoumarins. Numerous studies revealed that it has valuable pharmacological actions in peripheral tissues, such as anti-carcinogenic, anti-inflammatory, anti-Helicobacter, and anti-genotoxic ones [1,2]. We recently reported that (1) AUR can pass through the blood–brain barrier; (2) it exerts anti-inflammatory effects in the brain as well as in peripheral tissues, consequently exerting neuroprotective effects in the brain; and (3) its neuroprotective effects might be mediated by the suppression of the inflammatory response, as it effectively suppresses hyper-activation of microglia/astrocytes and hyper-expression of cyclooxygenase-2 and pro-inflammatory cytokines [3,4,5,6]. On the other hand, the fact that astrocytes have the ability to support neurons by secreting neurotrophic factors such as glial cell line-derived neurotrophic factor (GDNF), brain-derived neurotrophic factor (BDNF), and nerve growth factor have been well documented [7]. GDNF, in particular, has been reported to inhibit microglial activation [8]. Therefore, the most important object of the present study was to investigate whether AUR would have the ability to induce GDNF expression in astrocytes. As astrocytes, we used rat C6 glioma cells (rat glial tumor cell line), which have been frequently and extensively used as an alternative to astrocytes in primary culture [9,10,11,12,13], because these cells have similarities with primary astrocytes in culture [14,15] and have ability to synthesize/secrete high levels of GDNF [9,10,11].

GDNF was originally found as a trophic agent for midbrain dopamine neurons [16], and therefore the expectation was soon raised that it might be useful as a therapeutic agent to treat Parkinson’s disease (PD) [17]. In fact, many studies have shown that GDNF has the therapeutic benefits for various animal models of PD [18,19]. Later, it became clear that GDNF is expressed abundantly throughout the brain and that it also has a neuroprotective effect on noradrenergic neurons in the locus ceruleus of rodents [20,21,22].

In this study we successfully found that AUR had the ability to induce GDNF synthesis/secretion in C6 cells and then investigated the signal transduction molecule(s) at work that allowed AUR to induce *gdnf* gene expression.

## 2. Results

### 2.1. Effects of AUR on the Viability of C6 Cells

We initially evaluated the effect of 24 h-exposure to AUR on the cell viability. For this, C6 cells were seeded on a 96-well plate and cultured for 24 h in a medium containing 10% fetal bovine serum (FBS), and then treated with 10–80 μM AUR for 24 h in the same medium. Other cells on a 96-well plate were cultured for 24 h in a medium containing 10% FBS, and thereafter for another 24 h in medium containing 2% FBS. The cells were then treated with 10~80 μM AUR for 24 h in a medium containing 2% FBS. The results of MTT assay showed no differences in cell number between non-treated cells and those incubated with AUR (10–40 μM) both in medium containing 10% FBS (Figure 1A) and 2% FBS (Figure 1B). However, a decrease in cell viability was observed when the concentration of AUR was at or exceeded 60 μM in both medium. Based on these results, we chose 10–40 μM AUR for use in subsequent experiments. During the viability experiment, no apparent morphological changes (such as flattening and development of cell processes) were observed for cells treated at any of the concentrations tested.

### 2.2. Effects of AUR on GDNF Content of Conditioned Media

To examine the effect of AUR-treatment on the release of GDNF, we incubated C6 cells with 10 μM AUR for 0~60 h. As shown in Figure 2A, a significant increase in GDNF release by AUR was detectable at 40 h (** *p* < 0.01), which release remained elevated up to 60 h (** *p* < 0.01). To assess the concentration-dependency of AUR on the release of GDNF from C6 cells, we treated the cells with 20 or 40 μM AUR for 40 h. As shown in Figure 2B, a significant increase in GDNF release (** *p* < 0.01) was detectable at either concentration. These results thus showed that AUR induced GDNF release in a time-dependent and dose-dependent manner.

### 2.3. Effects of AUR on GDNF Levels in Cell Lysates

To examine the effect of treatment with AUR on GDNF expression in C6 cells, we treated them with various concentrations (0, 10, 20, and 30 μM) of AUR for 40 h. The results of immunoblot analysis (Figure 3) showed that the GDNF content in the control cell lysate was low but that significant induction occurred after 40 h of treatment with 30 μM AUR (* *p* < 0.05).

### 2.4. Effects of AUR on GDNF mRNA Expression in C6 Cells

We next investigated the time course of GDNF mRNA expression in C6 cells. C6 cells were treated with 10 μM AUR for 0~60 h. Figure 4A shows that AUR started to enhance the GDNF mRNA expression at 20 h of treatment (** *p* < 0.01), and continued to increase it even up to 60 h. We then examined the dose-dependency of AUR on GDNF mRNA expression when the cells were treated with 5, 10, 20, or 40 μM AUR for 40 h. Figure 4B shows that GDNF mRNA significantly increased by AUR-treatment. These results showed that AUR enhanced GDNF mRNA expression in a time-dependent manner and in a somewhat dose-dependent one.

### 2.5. Effects of AUR on the Phosphorylation of CREB and ERK

The phosphorylation of cAMP response element-binding protein (CREB), a transcription factor located within the nucleus, has been reported to be associated with GDNF expression as well as that of BDNF [14,23]. We thus examined whether AUR could elicit phosphorylation of CREB in C6. When C6 cells were treated with 20 μM AUR for 40 h, the ratio of phosphorylated CREB (pCREB) to total CREB (CREB) was significantly (** *p* < 0.01) increased (Figure 5A).

As the signaling pathway involved in the regulation of GDNF expression, extracellular signal-regulated kinases 1/2 (ERK1/2), which are components of the mitogen-activated protein kinase (MAPK) signaling pathway, are well known [13]. Figure 5B shows that AUR increased the ratio of phosphorylated ERK2 (pERK2) to total ERK2 (ERK2). Among ERK isoforms (ERK1; 44 kDa and ERK2; 42 kDa), we analyzed the ratio of phosphorylated ERK2 (pERK2) to total ERK2 (ERK2), because a less important/dispensable role was reported for ERK1 [24]. The results shown in Figure 5B indicate that AUR directly and significantly (* *p* < 0.05) activated ERK1/2.

### 2.6. Effects of Inhibitors of ERK/PKA on AUR-Induced GDNF mRNA Expression in C6 Cells

To examine whether the phosphorylation of ERK caused by AUR was responsible for GDNF expression, we pretreated C6 cells with 10 μM U0126 (a specific inhibitor of MAPK/ERK kinase 1; MEK1) for 30 min and subsequently treated them with 20 μM AUR for 40 h. Figure 6 shows that AUR-induced GDNF mRNA expression was significantly (^##^
*p* < 0.01) inhibited by U0126, indicating that the action of AUR on GDNF synthesis in astrocytes was mediated in an ERK/CREB-dependent manner.

In PC12 cells, the Ca^2+^-induced nuclear translocation of ERK to the nucleus requires protein kinase A (PKA) activation [25]. So we pretreated C6 cells with 1 μM H89 (a specific inhibitor of PKA) for 30 min and subsequently treated them with 20 μM AUR for 40 h. Figure 6 shows that AUR-induced GDNF mRNA expression was significantly (^#^
*p* < 0.05) inhibited by H89, indicating that the action of AUR on GDNF synthesis in astrocytes was mediated in an ERK/PKA/CREB-dependent manner. Figure 6 also shows that there was no effect of either inhibitor tested alone on the basal level of GDNF mRNA.

## 3. Discussion

The present study revealed that AUR had the ability to induce GDNF release in C6 cells (Figure 2), and that the AUR-induced GDNF release was associated with increased expression of GDNF mRNA (Figure 4). In order to eliminate the possibility that this release of GDNF was due to a leakage of cells as a consequence of AUR’s toxicity or an increase of cell population as a consequence of AUR’s growth promoting ability, we treated cells with AUR at the concentration of 10~40 μM in the present experiments (Figure 1). In experiments using this concentration range, we did not observe any differences (viability, morphological change etc.) between none-treated cells and AUR-treated cells up to 60 h. On the other hands, recent study showed that AUR was cytotoxic against U87 cells, human malignant glioblastoma (GBM) cell line, dose- and time-dependently with IC_50_ values of 108.9 μg/mL (namely, 0.365 μM) and 79.17 μg/mL (namely, 0.266 μM) obtained for 24 and 48 h-treatments [26]. Considering these results and our findings suggests the possibility that AUR exerts reversible effects on U87 cells, devastating type of astrocytic tumor, and C6 cells, astrocytoma cells that resemble normal astrocytes.

We also found presently that (1) AUR had the ability to directly elicit phosphorylation of ERK1/2 and CREB (Figure 5); (2) the AUR-induced GDNF mRNA expression was significantly inhibited by U0126, a MEK1 inhibitor, and H89, a PKA inhibitor (Figure 6), suggesting that AUR induced GDNF expression was mediated by the MAPK/ERK signaling pathway and PKA activation. As the mechanism underlying the upregulation of GDNF expression, the PI3K/Akt pathway [27] and PKC pathway [28] have also been reported to be involved. As far as we examined, neither the direct phosphorylation of Akt in response to AUR nor the inhibition of induction of GDNF by LY294002 (a specific inhibitor of PI3K) was observed (data not shown). The involvement of the PKC pathway on GDNF induction is yet still uncertain. In any event, the findings here suggest that AUR could cause the induction of GDNF expression in astrocytes, at least partly, by activating the PKA/ERK/CREB pathway.

We previously reported that AUR exerts powerful neuroprotective effects in vivo in the brain, probably because it suppresses inflammatory responses in the brain of a mouse model of global cerebral ischemia [3,5] and in lipopolysaccharide (LPS)-induced systemic inflammatory model mice [4,6]. In addition, we recently reported that the peel of *C. kawachiensis*, a rich source of AUR, is neuroprotective in LPS-induced systemic inflammatory model mice [4], senescence-accelerated mice [29], global cerebral ischemia model mice [30], type-2 diabetic db/db mice [31], and an LPS-induced mouse model of PD [32]. We also showed that the AUR-rich fruit juice of *C. kawachiensis*, made by the addition of peel paste to the raw juice, significantly suppressed the ischemia-induced neuronal cell death in the hippocampus of ischemic mice [33] and that this juice contributed to the preservation of cognitive function of healthy volunteers [34]. In these reports, we confirmed that the basis of this neuroprotective ability might have been anti-inflammatory effect of AUR. The findings obtained presently might offer a novel mechanism through which AUR and the peel of *C. kawachiensis* may exert their neuroprotective effects, i.e., the induction of GDNF in concert with the suppression of inflammation. In other words, the present findings suggest that the induction of GDNF might be another part of the mechanism accounting for the neuroprotective effects of AUR and the peel of *C. kawachiensis*. We are planning to confirm the AUR-induced GDNF expression in vivo using various disease model mice.

Accumulating recent studies showed that GDNF has the ability to improve brain function and that it plays an important role in various neuropsychiatric disorders such as depression, Alzheimer’s disease (AD), and aging [35]. As for AD, (1) the administration of GDNF protects against aluminum-induced apoptosis in rabbits by up-regulating Bcl-2 and Bcl-XL and inhibiting mitochondrial Bax translocation [36]; (2) Lenti-GDNF gene therapy protects against AD-like neuropathology in 3xTg-AD mice [37]; and (3) the level of mature GDNF peptide is down-regulated in the postmortem middle temporal gyrus of AD patients [38]. A recent review stated that various phytochemicals, including bioactive compounds in food and beverages, can protect neuronal cells by neurotrophic-factor mimetic activities in the brain [39]. The present findings taken together with the above findings supports our proposition [34] that AUR-rich *C. kawachiensis* peels might be a functional food for aging and neurodegenerative disorders.

## 4. Materials and Methods

### 4.1. Chemical and Reagents

AUR was kindly supplied by Ushio ChemiX Corp. (Omaezaki, Shizuoka, Japan). U0126 and H89 were purchased from Calbiochem Corp. (San Diego, CA, USA). These compounds were dissolved in dimethyl sulfoxide (DMSO). The final concentration of DMSO in all culture media was below 0.1%.

### 4.2. Cell Culture

All cell culture materials such as Dulbecco’s modified Eagle medium (DMEM), FBS, and antibiotics were purchased from Thermo Fisher Scientific (Waltham, MA, USA) as previously described [13]. C6 cells were maintained in a medium containing 10% FBS. For the MTT assay, cells were seeded in a 96-well plate at a density of 1 × 10^4^ cells/well. For other experiments, cells were seeded in 35-mm culture dishes at a density of 2.5 × 10^5^ cells/dish. Cells were cultured for 24 h in medium containing 10% FBS, and then for a further 24 h in medium containing 2% FBS, after which the cells were treated for the indicated times with samples in medium containing 2% FBS.

### 4.3. Determination of Cell Viability

Cell viability was determined by the use of the MTT assay, as previously described [40]. All assays were performed 4 times independently.

### 4.4. ELISA

C6 cells in 35-mm culture dishes were treated with AUR at the desired concentrations and times. The culture supernatants were centrifuged at 2000× *g* for 20 min to remove particulates before analysis. GDNF contents in cell culture supernatants were measured by using the GDNF E_max_^®^ ImmunoAssay System (Promega Corp., Madison, WI, USA).

### 4.5. Immunoblot Analysis

Cell extracts of C6 cells grown in 35-mm culture dishes were prepared as previously described [40]. Equal amounts of proteins (20 μg for pCREB and 10 μg for others) were separated on SDS-polyacrylamide gels and electroblotted onto an Immuno-Blot™ PVDF membrane (Bio-Rad Lab., Hercules, CA, USA). As primary antibodies, rabbit polyclonal antibodies against 44/42 ERK1/2 and phospho-44/42 MAPK (Thr202/Tyr204) were purchased from Cell Signaling Tech. Inc. (Woburn, MA, USA). Rabbit polyclonal antibodies against actin and GDNF were purchased from Sigma-Aldrich Company Ltd. (St. Louis, MO, USA) and Abcam plc. (Cambridge, Cambridgeshire, UK), respectively. As secondary antibody, alkaline phosphatase-linked anti-rabbit IgG (Cell Signaling) were used. Immunoreactive bands were detected by use of the NCB/BCIP reagent (Roche Diagnotics GmbH, Mannheim, Germany).

### 4.6. Total RNA Extraction and RT-PCR

C6 cells in 35-mm culture dishes were treated with AUR at the desired concentrations and times. Total RNA was prepared from the cells by use of Isogen II, a reagent for extracting total RNA (Nippon Gene Co., Ltd., Tokyo, Japan) and transcribed into cDNA by using a PrimeScript™ RT-PCR Kit (Takara Bio. Inc., Kusatsu, Shiga, Japan). The synthesized cDNA was amplified by PCR using each primer pair. The following primer pairs were used: 5’-GAG AGG AAT CGG CAG GCT GCA GCT G-3’ (25 mer) and 5’-CAG ATA CAT CCA CAT CGT TTA GCG G-3’ (25 mer) for GDNF; 5’-CGG AGT CAA CGG ATT TGG TCG TAT-3’ (24 mer) and 5’-AGC CTT CTC CAT GGT GGT GGA GAC-3’ (24 mer) for glyceraldehyde-3-phosphate dehydrogenase (GAPDH). The number of PCR cycles and specific annealing temperatures were 34 cycles and 60 °C for GDNF, and 26 cycles and 55 °C for GAPDH. Reaction products (337 bp for GDNF and 309 bp for GAPDH) were electrophoresed on 2% agarose gels containing ethidium bromide and photographed under UV light.

### 4.7. Statistical Analysis

All results were expressed as means ± SEM. Significant differences in experiments with 2 groups were analyzed by performing Student’s *t*-test. Experiments with 3 or more groups were subjected to a one-way ANOVA followed by Dunnet’s multiple comparison test. *p* < 0.05 was taken to be statistically significant.

## Figures and Tables

**Figure 1 ijms-21-00253-f001:**
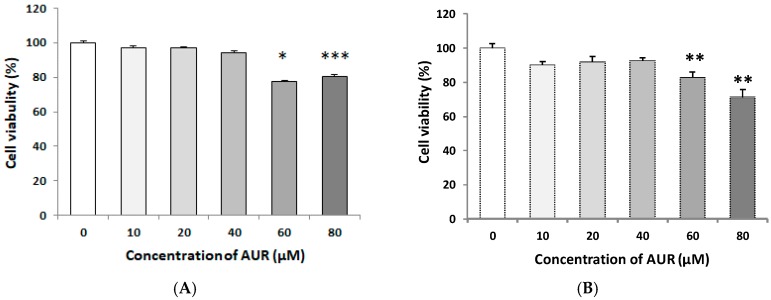
Effects of treatment with auraptene (AUR) on C6 cell viability in medium containing 10% fetal bovine serum (FBS) (**A**) or 2% FBS (**B**). Cells were treated with various concentrations (0–80 μM) of AUR for 24 h. The results are presented as the mean ± SEM (*n* = 4). Significance difference in values between the non-treated (0 μM) and AUR-treated cells: * *p* < 0.05; ** *p* < 0.01; *** *p* < 0.001.

**Figure 2 ijms-21-00253-f002:**
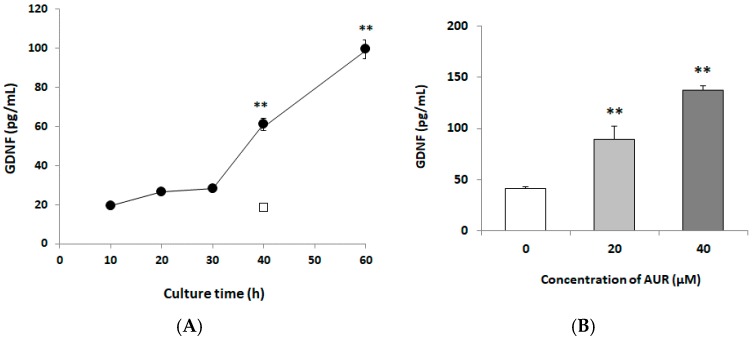
Effects of treatment with AUR on glial cell line-derived neurotrophic factor (GDNF) content in the medium of C6 cells. (**A**) Cells were incubated with 10 μM AUR for various times (10–60 h) (●) or without AUR for 40 h (□). Significance difference in values between the non-treated cells (40 h) and other cells: ** *p* < 0.01; (**B**) Cells were incubated with various concentrations (0, 20, and 40 μM) of AUR for 40 h. Significance difference in values between the non-treated (0 μM) and AUR-treated cells: ** *p* < 0.01. In (**A**) and (**B**), the results are presented as the mean ± SEM (*n* = 4).

**Figure 3 ijms-21-00253-f003:**
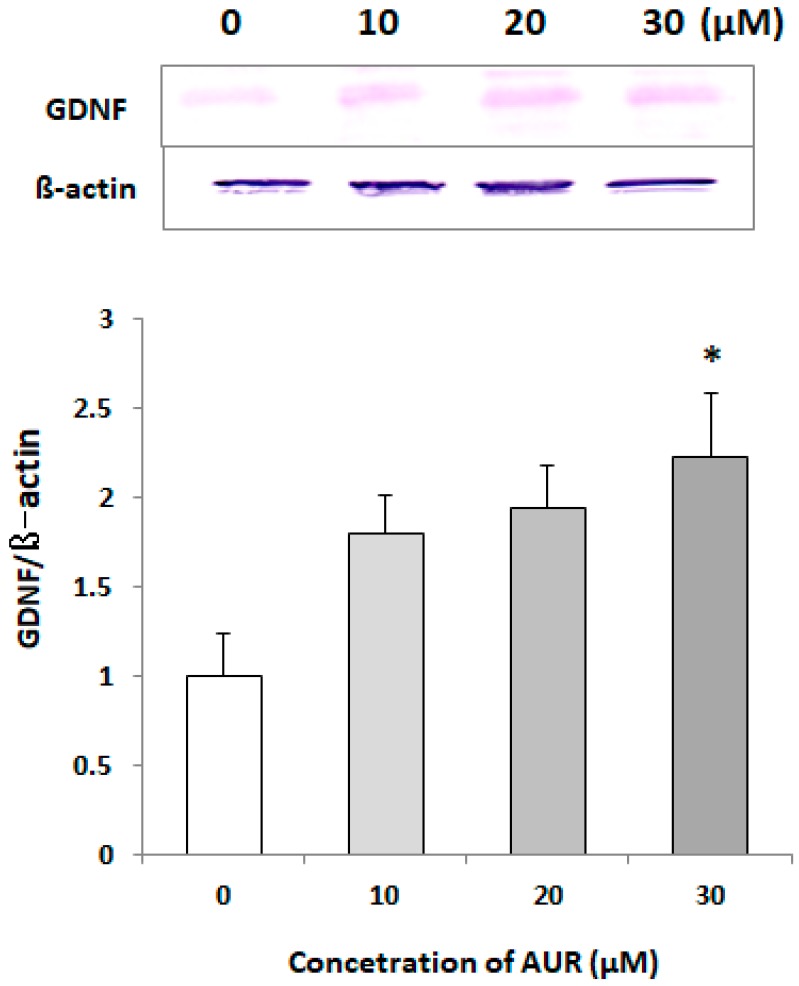
Effects of AUR-treatment with AUR on GDNF content in C6 cells. Cells were incubated with various concentrations (0, 10, 20, and 30 μM) of AUR for 50 h. The results are presented as the mean ± SEM (*n* = 3). Significance difference in values between the non-treated and AUR-treated cells: * *p* < 0.05.

**Figure 4 ijms-21-00253-f004:**
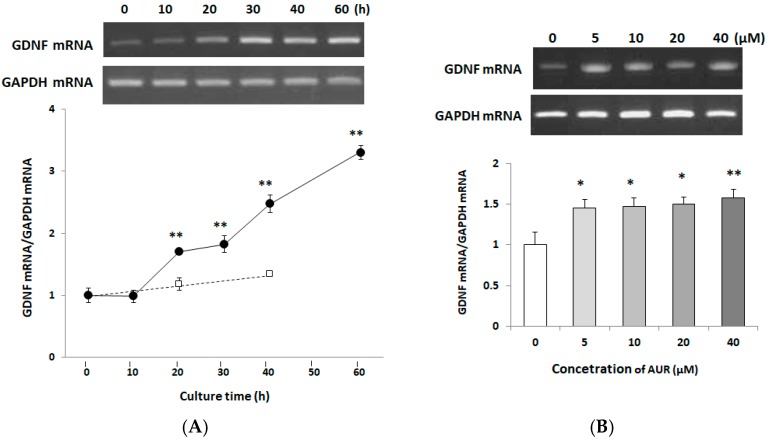
Effects of treatment with AUR on GDNF mRNA content in C6 cells. (**A**) Cells were incubated with 10 μM AUR for various times (0–60 h) (●) or without AUR for 20 or 40 h (□). Significance difference in values between the non-treated and AUR-treated cells: ** *p* < 0.01; (**B**) Cells were incubated with various concentrations (0, 5, 10, 20, and 40 μM) of AUR for 40 h. Significance difference in values between the none-treated (0 μM) and AUR-treated cells: * *p* < 0.05; ** *p* < 0.01. In (**A**) and (**B**), the results are presented as the mean ± SEM (*n* = 6).

**Figure 5 ijms-21-00253-f005:**
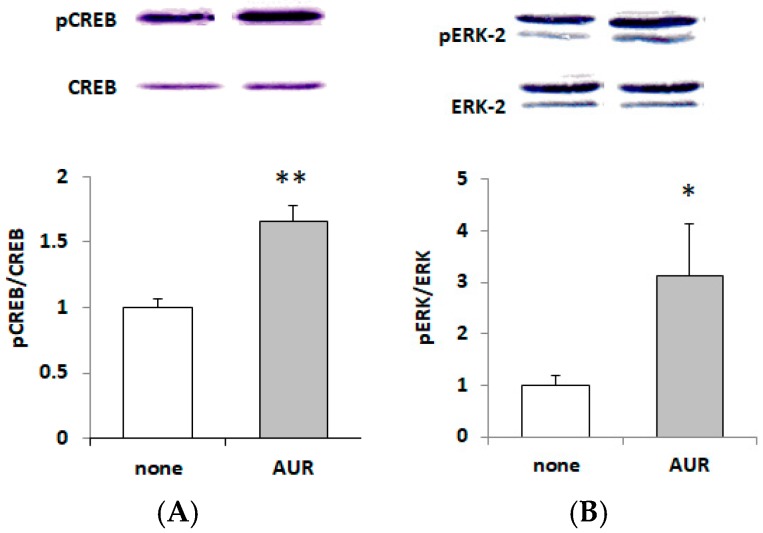
Effects of treatment with AUR on phosphorylation of cAMP response element-binding protein (CREB) (**A**) and extracellular signal-regulated kinase 2 (ERK2) (**B**) in C6 cells. Cells were incubated with 20 μM AUR for 40 h. The density ratio of phosphorylated components to total components of untreated cultures (none) was expressed as 1.0. The results are given as the mean ± SEM (*n* = 3). Significance difference in values between the none-treated and AUR-treated cells: * *p* < 0.05; ** *p* < 0.01.

**Figure 6 ijms-21-00253-f006:**
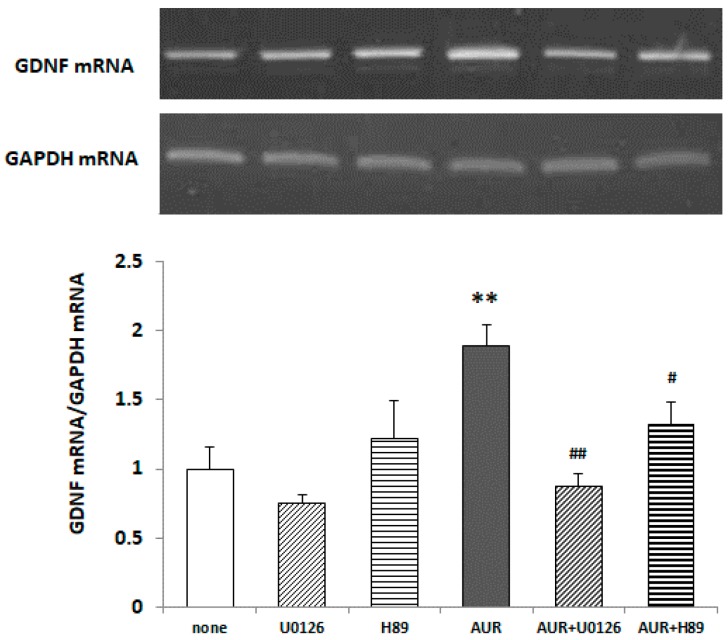
Effects of U0126 and H89 on AUR-induced GDNF mRNA content in C6 cells. Cells were preincubated with or without 10 μM U0126 or 1 μM H89 for 30 min and then incubated with 20 μM AUR for 40 h. The results are presented as the mean ± SEM (*n* = 4). Significance difference in values between the non-treated and AUR-treated cells: ** *p* < 0.05; significant difference in values between the AUR-treated and AUR/inhibitor-treated cells: ^#^
*p* < 0.05; ^##^
*p* < 0.01.

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
