# Peer review of "Citrus Auraptene Induces Glial Cell Line-Derived Neurotrophic Factor in C6 Cells"

_ijms, 2019, doi:10.3390/ijms21010253_

Round 1
Reviewer 1 Report
Two main points should be discussed by the authors:
Auraptene has been found elsewhere to be cytotoxic against a human malignant glioblastoma cell line. Although GDNF has originally been regarded as a trophic factor for the differentiation and survival of dopaminergic neurons in the embryonic midbrain, its overexpression has been suggested to yield oncogenic effects. GDNF mediates microglia attraction to tumour microenvironment, leading to improved tumour growth. GDNF has been reported to signal through multiple receptors to promote glioma development.The authors do not know the significance and potential consequences of an increase in GDNF supposedly induced by auraptene.
Perhaps auraptene might be a useful natural chemotherapeutic agent against gliomagenesis??
The Conclusions section is very simplistic and is no more than a synthesis of results.
Author Response
According to the kind suggestion of reviewer 1, we collected our manuscript as follows.
1) About “cytotoxic effect of auraptene”;
As pointed by reviewer, the cytotoxic effect of auraptene (AUR) against human malignant glioblastoma cell line (U87 cells) was recently reported. We cited this study as “reference 26” in the revised manuscript (line 220-222). But we have not observed the cytotoxic effect of AUR against C6 cells. We added Figure 1A (MTT assay under culture in 10% FBS-containing medium) and discussed about the possibility that AUR exerts reversible effects on U87 cells and C6 cells in the Discussion (line 222-225).
2) About “Conclusion is very simplistic and is no more than a synthesis of Results”;
As pointed by reviewer, the Conclusion of our previous manuscript was no more than a synthesis of Results. We omitted this section in our revised manuscript, as “Instructions for authors of IJMS” announced that “This section is not mandatory, but can be added to the manuscript if the discussion is unusually long or complex.”.
Reviewer 2 Report
In the manuscript entitled “Citrus auraptene induces glial cell line-derived neurotrophic factor in C6 cells”, Furukawa et. al showed that treatment of rat C6 glioma cells with AUR significantly increased in the expression and release of glial cell line-derived neurotrophic factor (GDNF), in a dose- and time-dependent manner. The authors further showed that the AUR-induced GDNF production was dependent on mitogen-activated protein kinase/extracellular signal-regulated kinase (ERK) 1/2, protein kinase A (PKA), and cAMP response element-binding protein (CREB). This study uncovers a potential mechanism of AUR in neuroprotection in some mouse models. However, there are several caveats of this study.
Major issues:
1) The authors need to polish the manuscript to be clearer and more precise. For example, in the Abstract “AUR phosphorylated cAMP response element-binding protein (CREB), a transcription factor located within the nucleus.” may be change to “AUR induced phosphorylation of cAMP response element-binding protein (CREB), a transcription factor located within the nucleus.”.
2) The C6 cell used in this manuscript is a glioma cell line. The authors may confirm the key finding with a primary glia cell culture.
3) Did the author check the C6 cell proliferation after AUR treatment?
4) The RT-PCR experiment in Figure 4 and 6 should have a H2O negative control for gdnf and gapdh gene.
5) The GDNF mRNA image in Figure 6 is not appropriate. Please replace it with a different image.
Minor issues:
1) The gene nomenclature should be changed. For example, the rat mRNA should be labeled as gdnf.
Author Response
According to the kind suggestion of reviewer 2, we collected our manuscript as follows.
1) About the expression that “AUR phosphorylated cAMP response element-binding protein (CREB), a transcription factor located within the nucleus.”
According to the kind advice of reviewer 2, we changed this sentence to “AUR induced phosphorylation of cAMP response element-binding protein (CREB), a transcription factor located within the nucleus.” (line 21-22).
2) About “The authors may confirm the key finding with a primary glia cell culture.”;
We added the explanation to explicitly show that C6 cells are frequently and extensively used as an alternative to astrocytes in primary culture (line 42-45) and increased the cited references [14-15]. As pointed by reviewer, it is important to confirm the key finding with a primary glia cell culture. We will confirm our present finding using primary glia cell culture.
3) About “Did the author check the C6 cell proliferation after AUR treatment? “;
As pointed by reviewer, it is important point whether AUR-induced GDNF expression is the result of the increase of the cells. As we have revealed that AUR treatment did not affect the proliferation of C6 cells, we added Figure 1A and discussed about this important point at Discussion (line 215-219).
4) About “H2O negative control for gdnf and gapdh gene for RT-PCR experiment”;
In the present study, we used classic semi-quantitative RT-PCR method. In this method, we understand that the measurement with the H2O negative control is not necessary.
5) About “The GDNF mRNA image in Figure 6 is not appropriate. Please replace it with a different image.”:
According to the suggestion of reviewer 2, we changed this image to another one.
6) About “This study uncovers a potential mechanism of AUR in neuroprotection in some mouse models.”;
As pointed by reviewer 2, we will confirm the findings observed by the present in vitro study using some pathological model mice. We explained our plan in Discussion (line 271-272).
7) About the gene nomenculture;
According to the suggestion of reviewer 2, we changed “GDNF gene expression” to “gdnf gene expression” (line 23 and line 55).
Round 2
Reviewer 1 Report
The manuscript deserves publication in IJMS
Author Response
Thank you for your kind help.
Reviewer 2 Report
The GDNF mRNA image in Figure 6 is still not appropriate. The last well in the replaced image still looks like not run the same experiment with other wells.
Author Response
The answer to Reviewer 2:
According to the kind suggestion of reviewer 2, we changed the image of Figure 6 to another one.